# Systematic dissection of key factors governing recombination outcomes by GCE-SCRaMbLE

Huiming Zhang[1,2,3,7], Xian Fu[2,3,4,7] ✉, Xuemei Gong[1,2,3], Yun Wang[2,3,4], Haolin Zhang[2,3], Yu Zhao[5] & Yue Shen[2,3,4,6] ✉

With the completion of Sc2.0 chromosomes, synthetic chromosome rearrangement and modification by loxP-mediated evolution (SCRaMbLE) becomes more critical for in-depth investigation of fundamental biological questions and screening of industrially valuable characteristics. Further applications, however, are hindered due to the lack of facile and tight regulation of the SCRaMbLE process, and limited understanding of key factors that may affect the rearrangement outcomes. Here we propose an approach to precisely regulate SCRaMbLE recombination in a dose-dependent manner using genetic code expansion (GCE) technology with low basal activity. By systematically analyzing 1380 derived strains and six yeast pools subjected to GCE-SCRaMbLE, we find that Cre enzyme abundance, genome ploidy and chromosome conformation play key roles in recombination frequencies and determine the SCRaMbLE outcomes. With these insights, the GCE-SCRaMbLE system will serve as a powerful tool in the future exploitation and optimization of the Sc2.0-related technologies.

Thanks to the rapid development of DNA sequencing and synthesis technologies, substantial progress has been made in the field of synthetic genomics, which enables the fundamental dissection of biological systems and offers potential solutions to many grand challenges[1,2]. In the past two decades, the attempts of building viral, bacterial and eukaryotic genomes have been achieved in succession[3]. As one of the milestone projects of synthetic genomics, the synthetic yeast genome project (Sc2.0) aims to build the first synthetic eukaryotic genome in the joint efforts of Sc2.0 international consortium[4–10]. As the unique built-in feature of Sc2.0 yeast genome, SCRaMbLE is designed to enable inducible whole-genome rearrangement by the Cre recombinase. Briefly, the symmetrical loxP (loxPsym) recombination sites that allow any pair of sites can recombine in either orientation[11], are inserted downstream of all non-essential genes in the synthetic

chromosomes[7]. When synthetic chromosomes are exposed to Cre recombinase, Cre-mediated recombination between two loxPsym sites leads to rearrangements of chromosome segments[12]. Owing to this notable feature, the synthetic yeast strains have been used as ideal chassis for both basic biological studies and engineering applications that are otherwise difficult to conduct[13–16].

While SCRaMbLE and associated analytical tools allow the generation and characterization of enormous population-level diversity in synthetic yeast cells[12], the potential of this technology is not fully unleashed due to the lack of systematic in-depth investigation of key factors affecting the SCRaMbLE process. In general, previous studies suggested the efficiency and recombination outcomes of SCRaMbLE are affected by several factors, including the abundance of functional Cre recombinase, genome ploidy, chromosomal conformation (liner

[1]College of Life Sciences, University of Chinese Academy of Sciences, Beijing 100049, China. [2]BGI Research-Shenzhen, BGI, Shenzhen 518083, China. [3]Guangdong Provincial Key Laboratory of Genome Read and Write, Shenzhen 518120, China. [4]BGI Research-Changzhou, BGI, Changzhou 213000, China. [5]Institute for Systems Genetics and Department of Biochemistry and Molecular Pharmacology, NYU Langone Health, New York, NY 10016, USA. [6]Shenzhen Institute of Synthetic Biology, Shenzhen Institutes of Advanced Technology, Chinese Academy of Sciences, Shenzhen 518055, China. [7]These authors contributed equally: Huiming Zhang, Xian Fu. ✉e-mail: fuxian1@genomics.cn; shenyue@genomics.cn

versus circular chromosomes), and the spatial proximity of loxPsym sites[12,15–18]. A recent investigation revealed that chromatin accessibility probability might also play an important role in the rearrangement outcomes by SCRaMbLE[19]. With the upcoming completion and consolidation of all Sc2.0 synthetic chromosomes, the complexity of SCRaMbLE outcomes will be inevitably increased. Thus, the systematic dissection of each factor with their direct and combined effects on the SCRaMbLE system becomes essentially necessary for further understanding and future extensive applications of this unique system.

Another challenge in studying the mechanism of SCRaMbLE and each contributing factor during the process is the limitations of current SCRaMbLE systems. The previously reported SCRaMbLE system is found to be relatively relaxed controlled due to the leaky activity of Cre recombinase and will lead to instability of rearrangement outcomes[20]. The light-controlled and AND gate strategies were developed to help control this process[17,20]. However, the light-controlled Cre-mediated recombination requires special equipment. The AND gate strategy relying on *GAL1* promoter is also not suitable for galactose utilization studies and it abandons the daughter-cell-specific *SCW11* promoter. The advantage of utilizing the *SCW11* promoter to disentangle factors involved in the SCRaMbLE process is crucial since it produces a pulse of recombinase activity only once in each cell's lifetime, resulting in relatively simple recombination events that are not too intricate to parse by sequence reconstruction. Thus, a facile method for tightly regulating the SCRaMbLE process while keeping the daughter-cell-specific promoter is desirable.

In this study, we present the GCE-SCRaMbLE system that exploits the genetic code expansion (GCE) technology to achieve tight regulation of Cre expression and to finely tune SCRaMbLE recombination frequency in a dose-dependent manner. Using synthetic yeast strains harboring various numbers of synthetic chromosomes, we detect no recombination event in the resulting SCRaMbLEd pools once GCE-SCRaMbLE is turned off. Using this system, we perform systematic chromosomal-rearrangement analysis using several synthetic yeast strains with distinct genome ploidy and chromosome conformation subjected to SCRaMbLE under different conditions. Compared to previous SCRaMbLE under selecting conditions, here, we perform a non-biased selection of SCRaMbLEd cells, aiming to reveal the real landscape of recombination events. By whole genome reconstruction and analysis of 1380 isolated SCRaMbLEd strains as well as SCRaMbLEd pools, our work reveals insights into the role of multiple factors affecting the SCRaMbLE outcomes, including the Cre recombinase abundance, the ploidy of hosting cells, and the conformation of the synthetic chromosome, which provides useful guidance on the future exploitation and optimization of the SCRaMbLE-mediated applications.

## Results

### GCE-SCRaMbLE allows tight control of Cre recombinase activity

The activity regulation of Cre enzyme by GCE was previously proposed with success using engineered pyrrolysyl-tRNA synthetase/tRNA$_{CUA}$ (PylRS/tRNA$_{CUA}$) pairs to encode photocaged amino acids[21,22]. However, the efficiency of these developed PylRS/tRNA$_{CUA}$ pairs is low in yeast[23]. Thus, we introduced an engineered leucyl-tRNA synthetase/tRNA pair (LeuOmeRS/tRNA$_{CUA}$) to the translational machinery of yeast cells to achieve efficient incorporation of *O*-methyl-*L*-tyrosine (OMeY) into the Cre recombinase in response to UAG stop codon[24]. The Cre enzyme is driven by the daughter-cell-specific promoter of *SCW11* to avoid on-going recombination events post-SCRaMbLE. In this way, the expression of full-length Cre recombinases depends on the presence of OMeY in cells (Fig. 1a). In a previous study, the estrogen-binding domain (EBD) was fused to the C-terminal of Cre recombinase to allow the estradiol-inducible regulation of SCRaMbLE[25]. However, the Cre recombinase functions as a tetrameric synaptic complex[26], and we suspect that the fusion of C-terminus EBD might affect its activity.

To test our hypothesis, we measured the recombination kinetics of purified Cre and Cre-EBD recombinases using an intramolecular excision assay[27]. Our result showed that the Cre-EBD exhibited 3.0-fold reduced affinity and 2.5-fold slower turnover rates compared with Cre without EBD, highlighting the presence of EBD does impair Cre enzyme activity (Supplementary Fig. 1). Therefore, the EBD was eliminated from our proposed GCE-SCRaMbLE system.

We next attempted to select a permissive residue of Cre enzyme for site-specific incorporation of OMeY. The N-terminal region of Cre recombinase was found to tolerate an insertion mutation[28]. Thus, codons corresponding to selected N-terminal residues (L5, L11, L14, and A18) were individually replaced with the UAG stop codon via site-directed mutagenesis. To test the performance of GCE-SCRaMbLE in yeast cells, a previously developed reporter system containing a loxP-flanked terminator cassette between the *ADH1* promoter and the CDS of the green fluorescent protein (GFP) reporter protein[17] was utilized to measure the recombination frequency (Fig. 1b). We transformed three plasmids encoding the LeuOmeRS/tRNA$_{CUA}$ pair, GFP reporter and Cre recombinase variants (Cre$_{UAG5}$, Cre$_{UAG11}$, Cre$_{UAG14}$, and Cre$_{UAG18}$) into yeast cells, followed by culturing cells in the presence and absence of 1 mM OMeY. Our data showed GCE-SCRaMbLE system could effectively trigger terminator deletion via recombination between two loxP sites induced by OMeY, leading to the detection of fluorescence signal. In addition, we observed Cre$_{UAG5}$-mediated GCE-SCRaMbLE was most efficient as the yeast cells expressing Cre$_{UAG5}$ exhibited the highest fluorescence intensity compared with other Cre variants. Consistent with the previous report[20], we found that the Cre-EBD system was indeed leaky: around 20% relative fluorescence signal was observed in the absence of β-estradiol. In contrast, our GCE-SCRaMbLE system showed minimal fluorescence signals at the background level in the absence of OMeY (Fig. 1b). Next, we investigated whether GCE-SCRaMbLE could be regulated by OMeY concentration. Using most active Cre recombinase variants (Cre$_{UAG5}$ and Cre$_{UAG14}$), we found the relative fluorescence intensity was positively correlated with OMeY concentration at 0, 0.5, 1, 2, 5, and 10 mM (Fig. 1c, d). Overall, we demonstrated GCE-SCRaMbLE exhibited low basal activity and allowed tight control of loxP-mediated recombination in a dose-dependent manner in our designed plasmid-based system.

### GCE-SCRaMbLE generates genome-wide recombination in synthetic yeast chromosomes

Taking advantage of the high controllability of GCE-SCRaMbLE system, we sought to determine the effect of Cre recombinase abundance, the conformation of synthetic chromosome, and genome ploidy on the SCRaMbLE outcome. A series of Sc2.0 yeast strains were used in this study: haploid yeasts harboring a single synthetic chromosome in both linear and circular form (*synII*, ring_*synII*) and multiple synthetic chromosomes (syn2369R: *synII*, *synIII*, *synVI*, and *synIXR*), heterozygous diploid yeast strains generated by mating the aforementioned haploid synthetic yeast strains with wild-type haploid yeast (BY4741 or BY4742). We transformed Cre recombinase variants (Cre$_{UAG5}$ and Cre$_{UAG14}$) and LeuOmeRS/tRNA$_{CUA}$ pair plasmids into these strains growing in the presence of OMeY to promote GCE-SCRaMbLE. In addition, four concentrations of OMeY at 1, 2, 5, and 10 mM were chosen for the induction using Cre$_{UAG14}$. In total, we specifically designed 23 test groups that are sufficient to fully dissect the influence and corresponding contribution of each factor during the SCRaMbLE process (Fig. 2a, Supplementary Table. 1).

SCRaMbLE occurrence rate was found to be positively correlated with cell death rate presumably because recombination events could lead to the deletion of essential genes[13], so assessment of SCRaMbLE-induced lethality is an easy and quick, although indirect, way to estimate the degree of genome rearrangements induced by SCRaMbLE. To determine the appropriate induction time, cultured cells of haploid strains induced by GCE-SCRaMbLE with 1 mM OMeY were plated at

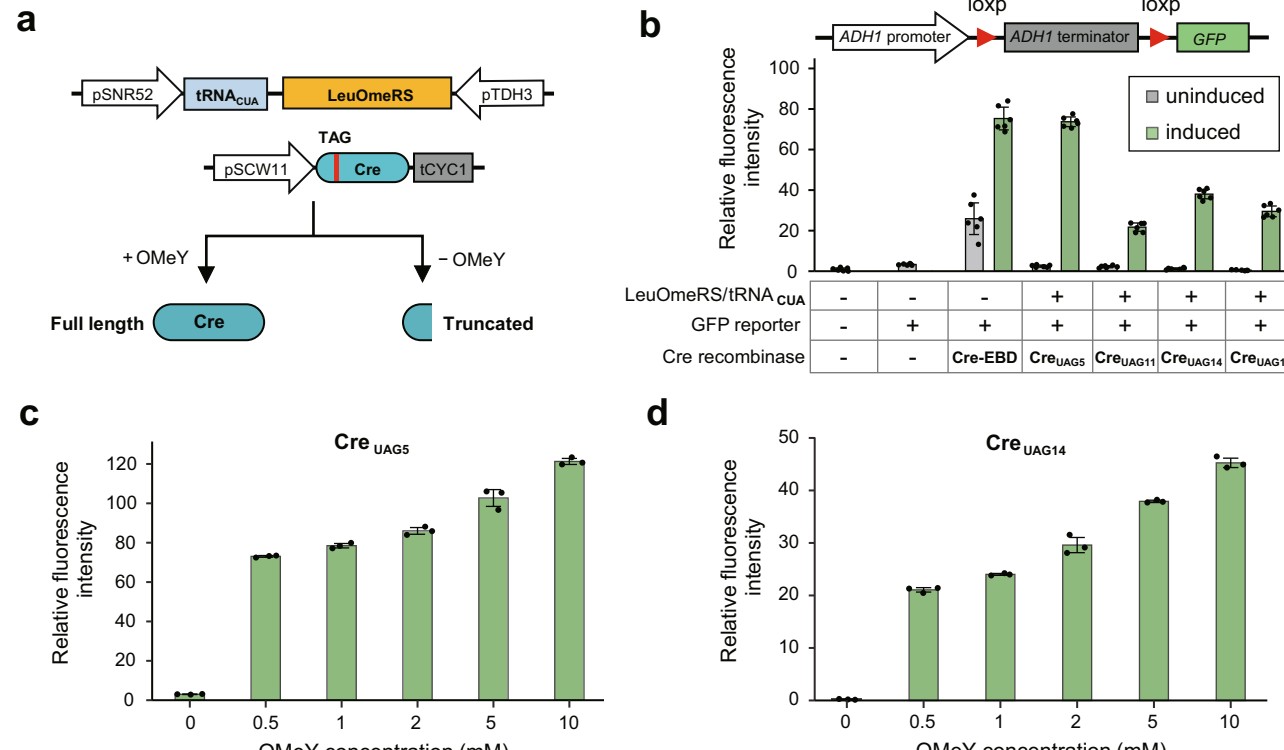

Fig. 1 | The GCE-SCRaMbLE system is designed for tight and tunable control of Cre recombinase expression. a Schematic representation of GCE-SCRaMbLE system, which utilizes the engineered leucyl-tRNA synthetase/tRNA (LeuOmeRS/tRNA$_{CUA}$) pair to incorporate OmeY into Cre recombinase in response to the TAG stop codon. The expression of full-length Cre enzyme depends on the presence of OmeY in the medium. b Measurement of recombination efficiency mediated by GCE-SCRaMbLE. Recombination efficiency was determined using a previously described reporter plasmid bearing a loxP-flanked ADH1 terminator cassette between ADH1 promoter and the coding region of GFP gene[17]. The loxP sites are shown as triangles in red. Wild-type yeast cells (BY4741) co-expressing LeuOmeRS/tRNA$_{CUA}$ pair, GFP reporter and Cre recombinase variants (Cre$_{UAG5}$, Cre$_{UAG11}$,

Cre$_{UAG14}$, and Cre$_{UAG18}$) were cultured in the presence (green) or absence (gray) of 1 mM OmeY for the calculation of recombination efficiency. For comparison, yeast cells co-expressing GFP reporter and Cre-EBD were induced with (green) or without (grey) 1 μM β-estradiol. c, d Measurement of relative fluorescence intensity of yeast cells co-expressing LeuOmeRS/tRNA$_{CUA}$ pair, GFP reporter and Cre$_{UAG5}$ (c) or Cre$_{UAG14}$ (d) cultured in the medium supplemented with varying concentrations of OmeY (0, 0.5, 1, 2, 5, and 10 mM). To calculate the relative fluorescence intensity, the fluorescence of yeast cells expressing GFP driven by ADH1 promoter serves as the reference. The error bars show the mean and standard deviation of multiple biological replicates (n = 6 for panel b, n = 3 for panel c-d). See Materials and Methods for details. Source data are provided as a Source Data file.

different time points for monitoring cell viability. We found cells exhibited the highest mortality after OmeY treatment for 24 h for most haploid strains (Supplementary Fig. 2). Thus, to acquire strains with SCRaMbLE events as much as possible when no selection was introduced, the induction time at 24 h was chosen for this study. To prevent early events that might dominate SCRaMbLE recombination outcomes, we performed 30 independent SCRaMbLE inductions in each group without selective pressures. For each induction, two colonies were randomly chosen post-recovery. To this end, 1380 (=23 × 30 × 2) GCE-SCRaMbLEd yeast strains were selected for SCRaMbLEd genome reconstruction and rearrangement analysis. In parallel with GCE-SCRaMbLE induction, four yeast pools (haploid syn2369R and ring-synII strains expressing Cre$_{UAG5}$ and Cre$_{UAG14}$ recombinase variants, respectively) in the absence of OmeY were also prepared for deep sequencing (-17,000×) to identify potential "leaky" events that might be overlooked by single colony selection.

We confirmed that no recombination was observed on synthetic chromosomes when designed yeast pools were cultured in the absence of OmeY, highlighting the very low basal activity of GCE-SCRaMbLE in its uninduced state. By whole-genome reconstruction of all 1380 GCE-SCRaMbLEd yeast strains, we observed SCRaMbLE events in 29.64% of the strain collection. In total, 1731 rearrangement events were identified in SCRaMbLEd cells. For strains with at least one SCRaMbLE event, our data revealed that the average number of SCRaMbLE events per strain is around 4. No inter-chromosomal events

were observed even though the quadruple-synthetic strain was utilized. Among all recombination events that occurred in cells, we observed three types of events, including inversion, deletion, and duplication, with the number of events for each type at 701, 765, and 265, respectively (Fig. 2b). To estimate the efficacy of GCE-SCRaMbLE to generate rearrangements through random recombination between two loxPsym sites, we built a rearrangement landscape by mapping junctions involving distinct recombination events to each loxPsym site on different synthetic chromosomes. We observed the widespread occurrence of GCE-SCRaMbLE-mediated recombinations in all synthetic chromosomes (Fig. 2c). The synII chromosome showed a higher number of recombination events than the other chromosomes because all testing strains contain this chromosome. Taken together, our data demonstrated that GCE-SCRaMbLE was able to generate synthetic yeast derivatives with genome-wide rearrangements.

## Recombination frequency correlates positively with Cre enzyme abundance regulated by GCE-SCRaMbLE

As GCE-SCRaMbLE controls Cre recombinase synthesis by incorporating OmeY into the growing peptide during the translation stage, the cellular abundance of Cre was expected to be positively correlated with the OmeY concentration. We performed quantitative western blot analysis to measure the cellular abundance of Cre variants (Cre$_{UAG5}$ and Cre$_{UAG14}$) with a gradient of OmeY concentrations. As expected, the cellular abundance of Cre$_{UAG14}$ was perfectly correlated with the

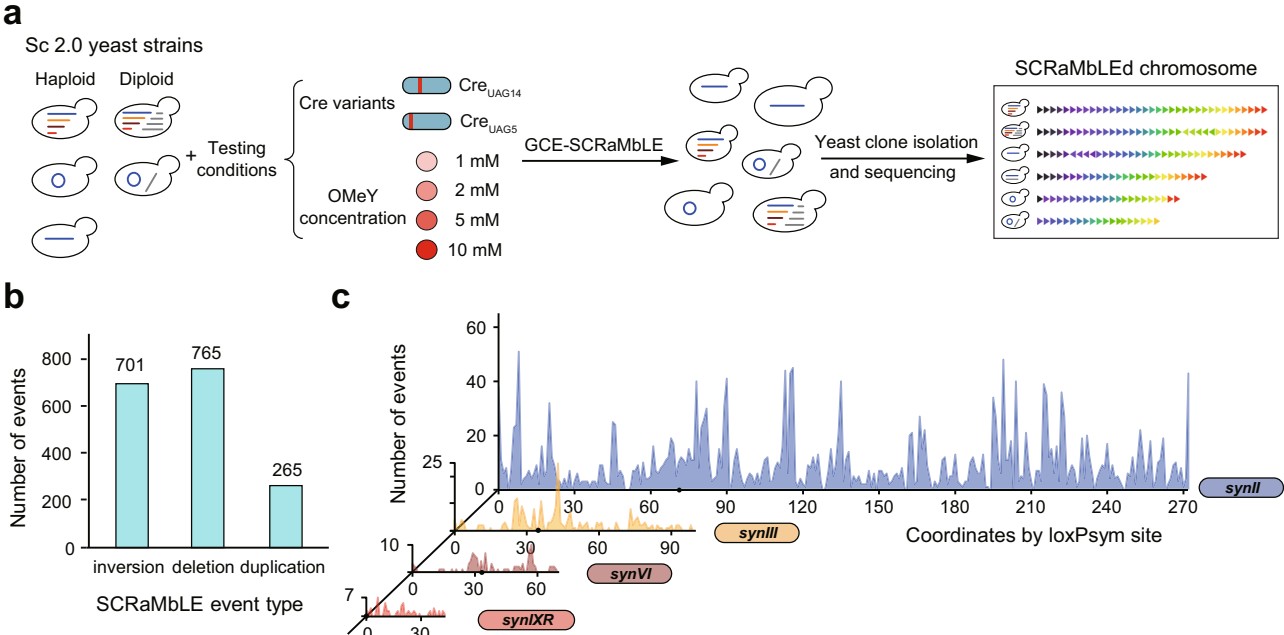

**a**

Sc 2.0 yeast strains

**Fig. 2 | The experiment design and resulting rearrangement landscape via GCE-SCRaMbLE. a** Schematic diagram showing the workflow of GCE-SCRaMbLE. A series of Sc2.0 yeast strains were used in this study: haploid and diploid yeasts harboring a *synII*, a ring_*synII* (the circular form of *synII*) and multiple synthetic chromosomes (*synII*, *synIII*, *synVI*, and *synIXR*). Testing conditions included four concentrations of OMeY (1, 2, 5 and 10 mM) for Cre_UAG14_ and one concentration of OMeY (1 mM) for Cre_UAG5_, which led to a total 23 groups of synthetic yeast strains subject to GCE-SCRaMbLE (see supplemental Table 1 for the design details). 60 colonies of SCRaMbLEd strains were randomly picked up from each group for whole genome sequencing and analysis. The SCRaMbLEd chromosome is represented as a sequence of arrows in which the color of each arrow indicates the different segments flanked by loxPsym recombination sites, and the direction of the arrow represents the orientation. **b** A histogram showing the number of different types of SCRaMbLE events, including inversion, deletion, and duplication. **c** Rearrangement landscape of four synthetic chromosomes that are present in all synthetic yeast strains. The number of events is calculated by mapping junctions involved in the rearrangement events to each loxPsym site. In the *synII* (blue) rearrangement landscape, all events observed in linear *synII* and circular *synII* were combined for analysis. The number of synthetic yeast cells bearing *synII*, ring_*synII*, *synIII* (yellow), *synVI* (brown), and *synIXR* (red) for analysis are 840, 540, 540, 540, and 540, respectively. Source data are provided as a Source Data file.

OMeY concentration (Fig. 3a). Consistent with our finding that the efficiency of GCE-SCRaMbLE mediated by Cre_UAG5_ was higher than that of Cre_UAG14_ (Fig. 1b), we also observed a higher abundance of Cre_UAG5_ than that of Cre_UAG14_ in yeast cells grown in the medium supplemented with 1 mM OMeY (Fig. 3a). In addition, compared with Cre_UAG5_, we observed no readthrough of the amber codon in the gene encoding Cre_UAG14_ without OMeY, demonstrating GCE-SCRaMbLE using Cre_UAG14_ has very low basal expression in the uninduced state.

We next aimed to explore whether and how the expression level of the Cre recombinase affects SCRaMbLE recombination outcomes. The concentration of OMeY and the choice of Cre variants are both contributing factors to the expression level of Cre recombinase. Here we investigated each factor and its influence in total of six subgroups of GCE-SCRaMbLEd strains, with different OMeY concentrations or Cre variants, in which 180 haploid strains per group were randomly selected for sequencing analysis. Similar to the results from the plasmid-based GFP reporter system, our result showed a positive correlation between the concentration of OMeY and the SCRaMbLE recombination frequency (Fig. 3b, c). The SCRaMbLE occurrence rate within a post-SCRaMbLE yeast population remained steady once the concentration of OMeY reached and exceeded 5 mM, suggesting the effective concentration of Cre enzyme became saturated. Strains with 8 and 11 recombination events were found when GCE-SCRaMbLE was induced by 5 and 10 mM OMeY, respectively. In contrast, the number of events was lower in the condition of 1 and 2 mM OmeY, at up to 3 and 4 events per strain, respectively (Fig. 3c). For the Cre variants, we found SCRaMbLE occurrence rate in the population and the number of SCRaMbLE events per cell were both higher in yeast cells expressing Cre_UAG5_ than that for Cre_UAG14_ (Fig. 3d, e), which could be explained by

the cellular abundance of these two Cre variants. We also tested whether the level of Cre enzyme played a role in the distributions of different recombination types, and no evident correlation was revealed (Supplementary Fig. 3). Our result suggests that the regulation of SCRaMbLE recombination frequency could be achieved by fine-tuning the concentration of OMeY and utilizing proper Cre recombinase variants.

## Genome ploidy is a key factor affecting deletion capability by SCRaMbLE and the resultant proportion of distinct types of events

Deletions of chromosome fragments bearing essential genes via SCRaMbLE could lead to the loss of viable cells in haploid strains, thus restricting the extent of deletable chromosome contents and potentially decreasing the genomic diversity post-SCRaMbLE. This issue could be addressed in the diploid background, allowing the synthetic chromosomes to undergo rearrangements while essential genes in native alleles remain unaffected. A recent study showed heterozygous diploid strains are more tolerant to genome rearrangements by SCRaMbLE compared to haploid strains[15]. To better understand the effect of genome ploidy on deletion capability via SCRaMbLE, we compared the loss of chromosome content on the target synthetic chromosome using 60 haploid strains and another corresponding 60 heterozygous diploid strains, subjected to the same GCE-SCRaMbLE induction. In general, the number of strains with reduced chromosome content and the degree of summative chromosome loss was significantly higher in the diploid background (Fig. 4a and Supplementary Fig. 4). For diploid strains, we found 19 strains showing reduced contents in the ring_*synII* with the lowest chromosome retention rate at

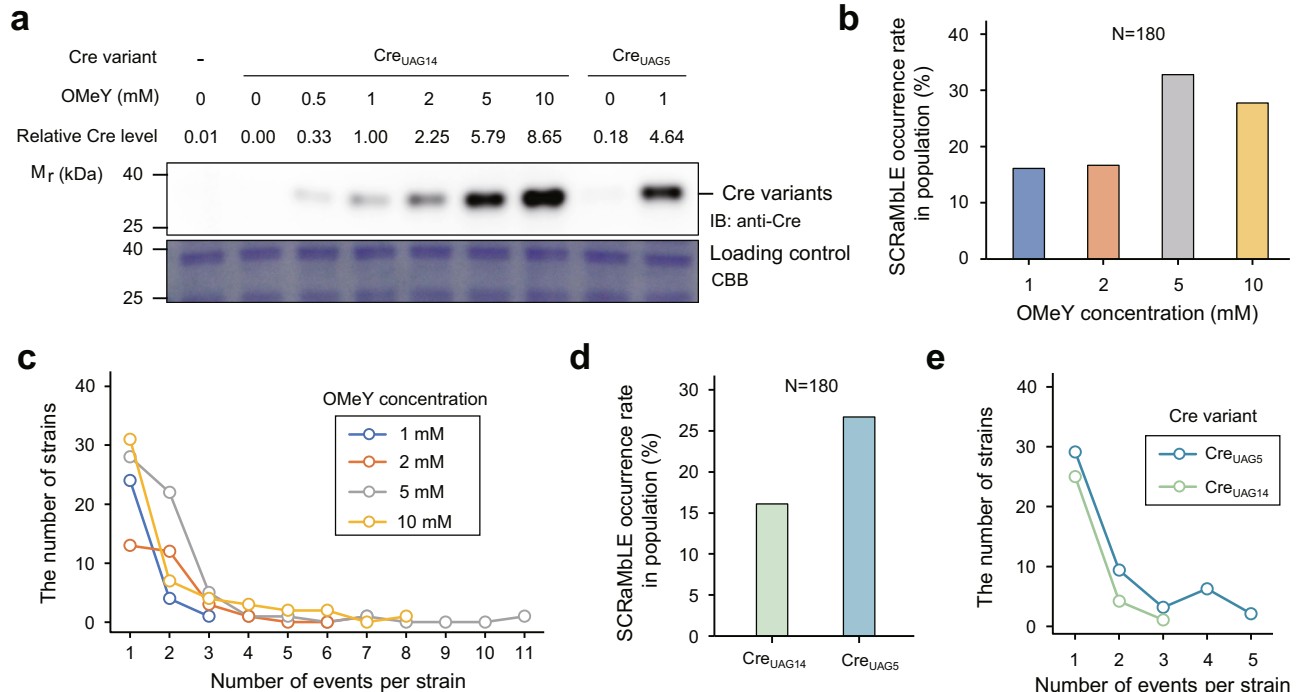

**Fig. 3 | The effect of Cre enzyme concentrations on SCRaMbLE outcomes.**
**a** Cellular abundance of Cre_UAG5 and Cre_UAG14 in yeast cells cultured in the medium with varying concentrations of OMeY (0, 0.5, 1, 2, 5, and 10 mM) was measured by western blot using an anti-Cre antibody. Relative abundances of Cre_UAG5 and Cre_UAG14 are indicated above the gel, and the cellular concentration of Cre_UAG14 produced by 1 mM OMeY is defined as 1.00 for comparison. Data were quantified by ImageJ (version 1.53e). Equal loading was confirmed by Coomassie brilliant blue (CBB) staining of parallel gels. Experiments were performed in triplicate, and representative images are shown. **b–e** The comparison of SCRaMbLE occurrence rate within a post-SCRaMbLE yeast population (**b** and **d**) and the number of all types of SCRaMbLE events per SCRaMbLEd strain (**c** and **e**) for Cre_UAG14-mediated GCE-SCRaMbLE induced by various concentrations of OMeY (1, 2, 5 and 10 mM) and for Cre_UAG5/Cre_UAG14-mediated GCE-SCRaMbLE induced by 1 mM OMeY, respectively. For these analyzes, a total of 180 haploid SCRaMbLEd strains bearing synII, ring-synII, and multiple synthetic chromosomes (synII, synIII, synVI, and synIXR) were used under each GCE-SCRaMbLE condition (different OMeY concentrations or Cre variants). Data referring to 1, 2, 5, and 10 mM OMeY concentration are labeled as blue, orange, grey and yellow, respectively. Data referring to Cre_UAG14 and Cre_UAG5 are labeled as light green and sky blue, respectively. Source data are provided as a Source Data file.

13.95%. In contrast, deletion events on the ring_synII were observed in only 13 haploid strains, with the chromosome retention rate higher than 98%. Similarly, we found 26 diploid cells have chromosome content loss (chromosome retention rate ranging from 29.37% to 99.96%) in linear synII after GCE-SCRaMbLE induction, while only 11 haploid cells with minimal loss of synII content less than 1% were observed. In addition, we did not observe the apparent effect of different Cre expressions on deletion capability by GCE-SCRaMbLE in the diploid strains (Supplementary Fig. 4).

To further characterize the genomic diversity between haploid and diploid SCRaMbLEd strains, we generated a rearrangement landscape by mapping all recombination loci to their corresponding loxPsym sites. We observed two notable effects of ploidy on the SCRaMbLE outcomes. First, the SCRaMbLE rearrangement frequency on the target chromosome synII was significantly higher across the entire chromosome in diploid strains than in haploid counterparts (Fig. 4b). The SCRaMbLE frequency for different regions of synII ranged from 7.92% to 17.08% in diploid strains, whilst in haploid strain, the average frequency of synII is as low as 1.11%. In addition, we noticed a significant difference in the proportion of distinct events between haploid and diploid cells (Fig. 4b). As the symmetry of loxPsym sites allows recombination in either orientation, deletions, and inversions were observed with approximately equal frequency in diploid strains as expected. In contrast, the proportion of deleted segments dropped significantly in haploid cells; and these deletion events were scattered sparsely through the synII, which could be explained by the distribution of essential genes on synII (Fig. 4b). The SCRaMbLE rearrangement landscape also revealed that no

regions in synII chromosome that are clearly prone to deletion, duplication, and inversion in the diploid strains except for the centromere region with relative low deletion events (Fig. 4b). Taken together, our findings highlight that ploidy has a significant impact on the type, diversity and proportion of SCRaMbLE-mediated rearrangement events.

**Circularization of synII leads to enhanced inter-chromosomal contacts and an increased number of SCRaMbLE events**
Chromosomes in circular conformation could lead to many complex rearrangement events after SCRaMbLE[12,18]. Here we further dissect its influence on the SCRaMbLE outcomes using strains carrying linear or circular synII in haploid or diploid background subjected to GCE-SCRaMbLE induction under the same condition. As expected, we found the average number of recombination events per SCRaMbLEd strain was much higher in cells bearing circular synII than that in the linear counterpart, regardless of genome ploidy. Specifically, the average numbers of recombination per SCRaMbLEd strain in haploid and diploid cells were, respectively, 1.12 and 2.67 for linear synII, versus 1.40 and 8.86 for circular synII.

The linear and circular synII share the almost identical sequence but have distinct chromosomal conformation. We next exploited Hi-C to investigate the difference in the 3D structure between the linear and circular synII. To allow the qualitative comparison, 2D contact maps (bin size at 10 kb) and 3D projections for both linear and circular synII were generated based on the ligation frequencies between DNA restriction fragments (Fig. 5a). In general, we found circularization of synII led to apparent changes in its overall structure. The circular synII exhibited increased intra-chromosomal contacts than the linear

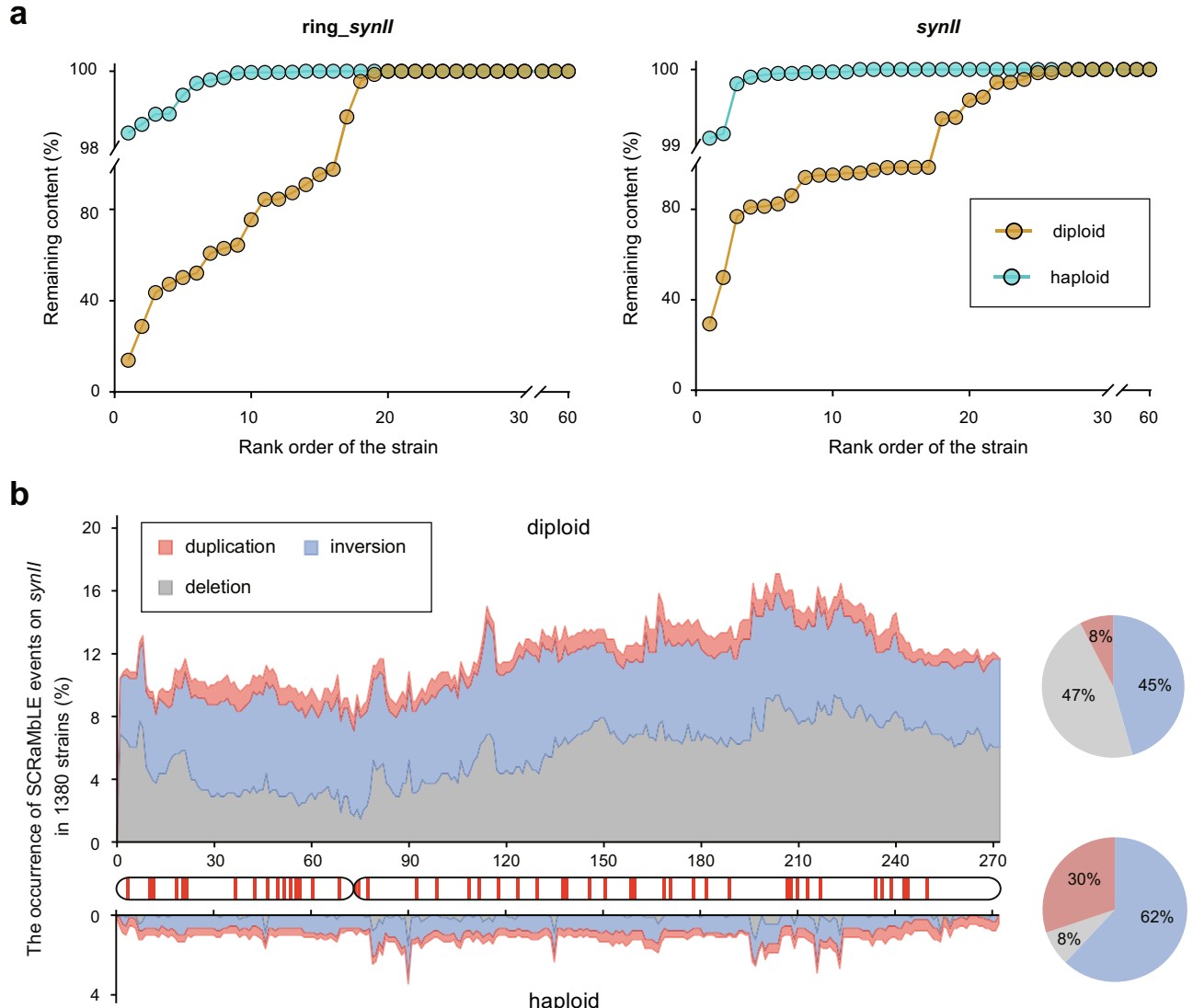

**Fig. 4 | The comparison of chromosome content loss and proportion of rearrangement types between haploid and diploid cells subjected to GCE-SCRaMbLE.** **a** The comparison of retention rate of synthetic chromosomes (*synII* and ring_*synII*) between diploid and haploid strains after GCE-SCRaMbLE under the same condition. 60 haploid and diploid yeast cells carrying ring_*synII* were respectively subjected to GCE-SCRaMbLE using Cre$_{UAG14}$ induced by 5 mM OMeY (left). 60 haploid and diploid yeast cells carrying 4 synthetic chromosomes (*synII*, *synIII*, *synVI*, *synIXR*) were respectively subjected to GCE-SCRaMbLE using Cre$_{UAG5}$ induced by 1 mM OMeY (right). Haploid and diploid strains are indicated with sky blue and golden yellow dots, respectively. **b** The comparison of *synII* SCRaMbLE

rearrangement landscape between 900 haploid (down) and 480 diploid (up) strains. The rearrangement landscape was generated by mapping various types of recombination occurring regions to *synII* referring to loxPsym sites. The *x*-axis refers to the number of loxPsym site (in total 272) on *synII* chromosome. Each section between every two loxPsym sites is indicated in red on the *synII* diagram if essential genes are present in the corresponding section. Duplication, inversion, and deletion are labeled as red, blue, and grey, respectively. Pie charts on the right shows the proportion of segments involved in the three different event types among all the rearranged chromosome segments identified in the 1380 isolated strains. Source data are provided as a Source Data file.

counterpart, especially between the two subtelomeric regions (30 kb length) at the chromosome ends, which are designated as regions of interest (ROI) in the contact maps and 3D representations (Fig. 5a). A previous study suggests 3D proximity of loxPsym sites on synthetic chromosomes is closely related to the likelihood of deletion event via SCRaMbLE[16]. We hypothesized that the increased intra-chromosomal contacts by circularization of *synII* chromosome would lead to increased occurrence of SCRaMbLE events in circular *synII* compared to linear *synII*. To test this, we performed ultra-deep sequencing (80,000×) to analyze genome rearrangements of two SCRaMbLEd cell populations that bear linear and circular *synII*, respectively. Only heterozygous diploid strains subjected to GCE-SCRaMbLE (1 mM OMeY) were utilized in the analysis to increase rearrangement frequency. Indeed, we observed more SCRaMbLE events in the circular *synII* than

that in the linear *synII* (10769 versus 7278). For quantitative analysis, we plotted the number of SCRaMbLE events against the normalized contact counts per bin between the regions involved in these recombination events. A significant positive correlation was revealed between the intra-chromosomal contact strength and the number of SCRaMbLE events (Fig. 5b). In addition, this positive correlation was stronger in circular *synII* compared to linear *synII*. Overall, these findings are consistent with our hypothesis that the frequency of SCRaMbLE events is positively correlated to the intensity of intra-chromosomal contacts.

Since the extra intra-chromosomal contacts in circular *synII* compared with linear *synII* mainly derived from the contacts between ROI and the rest part of the chromosome, we next investigated whether these ROI-dependent interactions were responsible

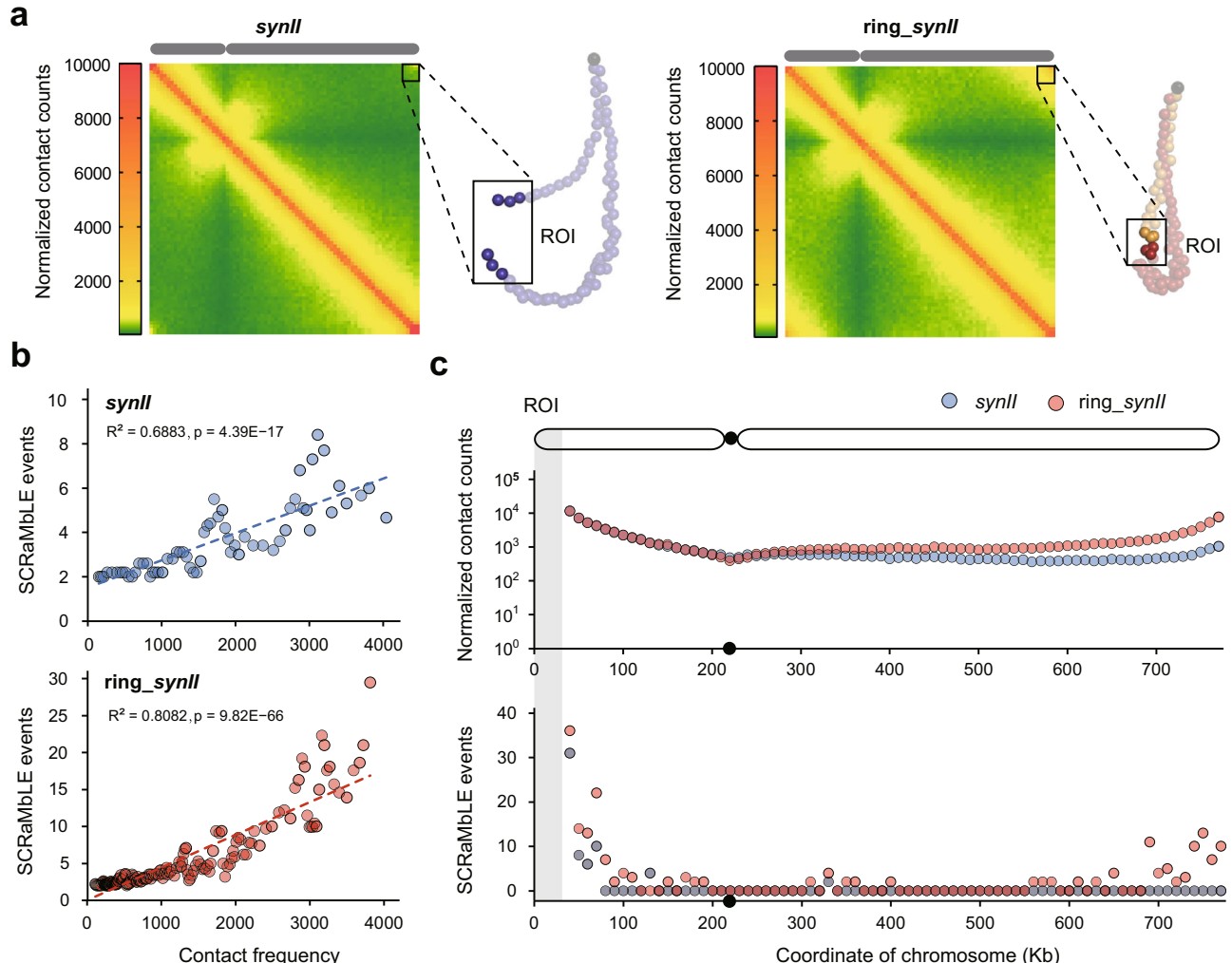

**Fig. 5 | The effect of chromosomal conformation on SCRaMbLE outcomes.**
**a** Contact maps (bin size at 10 kb) and corresponding 3D representations of *synII* and ring_*synII*. Normalized contact counts per bin are indicated, ranging from green (loose contacts) to orange (moderate contacts) and red (intense contacts). The right panels show 3D representations of the contact maps, and each bead represents a 10-kb chromosome segment. S*ynII*, left arm of ring_*synII*, and the right arm of ring_*synII* are indicated with blue, orange, and red dots, respectively. Two subtelomeric regions (30 kb length) designated as regions of interest (ROI) are shown and highlighted. **b** Correlation between the number of SCRaMbLE events and intrachromosomal contact strength in SCRaMbLEd cell populations carrying *synII* (up) and ring_*synII* (down). The least squares method was utilized to fit the linear regression to the data. R indicates Pearson's correlation. The *p*-value was calculated two-sided. Each dot represents the moving average of SCRaMbLE events over a varying contact counts window containing 10 data points, shifting the window by 3 data points per calculation. **c** Quantitative analysis of ROI-dependent intra-chromosomal contacts (up) and SCRaMbLE events (bottom) in *synII* (blue dots) and circular *synII* (red dots) by using ROI as the bait. The *y*-axis shows normalized contact counts per bin (10 kb) or the number of SCRaMbLE events; the *x*-axis shows the position (in kilobases) from the left arm to the right arm. See Materials and Methods for details. Source data are provided as a Source Data file.

for the increased number of SCRaMbLE events in circular *synII*. We focused on the intra-chromosomal contacts between the ROI in the left arm and other chromosome regions in linear *synII* or circular *synII* by means of a bait chromosome capture approach[29]. The contact counts between the ROI and every 10 kb region in the right arm of *synII* appeared to be higher in the circular form than that in the linear counterpart, and the contact discrepancies increased from centromere to the chromosome end (Fig. 5c). In a similar manner, we also compared the number of recombination events specifically occurred between the ROI and the rest part of linear and circular *synII*. We found the number of ROI-involved SCRaMbLE events was much higher in circular *synII* than that in the linear *synII*, especially at the telomere adjacent region of the right arm (Fig. 5c). Taken together, our results showed increased intra-chromosomal contacts, mainly from the interactions between the ROI and the other chromosomal parts in *synII*, gave rise to more ROI-dependent rearrangement events.

## Discussion

In this study, we demonstrated that the GCE-SCRaMbLE is a facile method to precisely regulate Cre recombinase activity by site-specific incorporation of OMeY, an unnatural amino acid that is specifically recognized by LeuOmeRS and rejected by the endogenous aminoacyl-tRNA synthetases, into the Cre enzyme in response to the amber stop codon. This method holds two advantages: low background activity and flexible adjustability. Minimizing the uninduced activity of Cre is essential to maintain Sc2.0 chromosome stability. Our results indicated that GCE-SCRaMbLE exhibits undetectable basal activity based on the GFP-based fluorescence assay. Deep sequencing of yeast pools subjected to GCE-SCRaMbLE in the absence of OMeY further confirmed that no SCRaMbLE event occurred on the synthetic chromosomes. The low basal activity of GCE-SCRaMbLE is due to the minimal stop-codon readthrough of Cre$_{UAG14}$ in the absence of OMeY, indicating the high orthogonality of the introduced LeuOmeRS/tRNA$_{CUA}$ pair and low amber suppression level in the budding yeast. Previous

works have shown that the expression of Cre recombinase could be regulated at transcriptional and post-translational levels[17,20,25,30]. Here we further demonstrated that SCRaMbLE recombination frequency can be fine-tuned by regulation of Cre expression at the translational level. The protein abundance was able to be easily altered either by using different Cre variants containing in-frame stop codons at distinct positions or by manipulating the concentration of OMeY. The different expression levels of Cre variants could be attributed to the mRNA context effects of GCE by amber codon suppression[31]. The capability of GCE-SCRaMbLE to regulate the abundance of Cre enzyme enables the direct investigation of the relationship between the Cre protein expression level and SCRaMbLE recombination outcomes.

Recombination frequency is a main driving force for rapid genome evolution via SCRaMbLE. To the best of our knowledge, the SCRaMbLE occurrence rate in the population scale and the number of events per strain without selection pressure have not been systematically investigated previously. By deep sequencing of 1380 randomly isolated yeast clones subjected to GCE-SCRaMbLE system, we found that about thirty percent of the strains underwent recombination events on the synthetic chromosomes. Although we observed the widespread occurrence of genomic rearrangements in all four synthetic chromosomes, the SCRaMbLE recombination frequency seemed to be relatively low. Thus, our finding highlights the importance of applying selective pressure or phenotypic screening to identify cells with genomic rearrangements subjected to the SCRaMbLE system for a broader range of applications, as demonstrated in previous studies[32,33]. We also envision that the recombination frequency mediated by GCE-SCRaMbLE would be further enhanced by improving the suboptimal incorporation efficiency of OMeY into Cre enzyme via the engineering of LeuOmeRS/tRNA$_{CUA}$ pair[34] and eukaryotic release factor 1 (eRF1) that competes with unnatural amino acid insertion in response to stop codons[35].

We confirmed that genome ploidy played a key role in determining SCRaMbLE outcomes, which is in agreement with previous studies using diploids for SCRaMbLE[15]. The SCRaMbLE events involving deleterious deletions were much more frequently observed in heterozygous diploids compared with haploids. With backup chromosomes to maintain the core functions of life, diploids are tolerant to deletions involving essential genes and therefore allowing much more beneficial genome reorganizations to occur for a wide range of purposes. In addition, we found that the circularization of the synthetic chromosomes will further promote the recombination frequency. By ultra-deep sequencing of SCRaMbLEd pools, our data revealed the recombination frequency in yeast pools with circular *synII* was increased by 30% in comparison with that carrying linear *synII*. The subtelomeric regions were found to be involved in many of the additional SCRaMbLE events. It is not surprising since the major difference between circular and linear *synII* is that the physical distance between the two subtelomeric regions is significantly decreased by circularization. Translocations induced by SCRaMbLE system have been identified in a previous study[29]. However, such inter-chromosomal/translocation events were not observed in our study, even with a relatively large sampling size (540 post-SCRaMbLE strains carrying four synthetic chromosomes). Considering that the inter-chromosomal interactions are affected by spatial distance between chromosomes[36], we speculate that the frequency of translocation events would be increased using the final Sc2.0 yeast strains harboring all synthetic chromosomes for SCRaMbLE applications. Apart from that, we envision that the extension of induction time, as suggested in the previous study[29], could also lead to an increased frequency of inter-chromosomal rearrangements.

Looking forward, the final Sc2.0 strain, in which all known TAG codons are substituted by TAA, will pave the way for the complete reassignment of amber-stop codons to unnatural amino acids and allow for many exciting applications in the full synthetic yeast[37]. By integrating genes encoding components of the GCE-SCRaMbLE system, including LeuOmeRS/tRNA$_{CUA}$ pair and Cre$_{UAG5}$, into the synthetic strain, our GCE-SCRaMbLE method is a perfect fit with the upcoming final Sc2.0 strain. The SCRaMbLE process could be turned on and off at will by simply switching the growth conditions (with or without OMeY), which will facilitate rapid and iterative genome evolution of Sc2.0 strain for many applications.

## Methods

### Strains and media
Yeast strains used in this study are described in Supplementary Table 2. Yeast strains were cultured at 30 °C in yeast extract peptone dextrose (YPD) (10 g/l yeast extract, 20 g/l peptone, 2% (w/v) glucose) or synthetic complete (SC) medium with two or three amino acids dropped out to select for the maintenance of plasmids carrying corresponding auxotrophic markers. The synthetic yeast strains used for genome rearrangement via GCE-SCRaMbLE were grown in SC−His−Leu medium supplemented with various concentrations of OMeY (Sigma-Aldrich, Cat number: 158259). The synthetic yeast strains used for fluorescence assay were grown in SC−His−Leu−Ura medium supplemented with OMeY or β-estradiol (Toronto research chemicals, Cat number: E888000). The isolated SCRaMbLEd strains for genome extraction and sequencing were grown in YPD to stationary phase. The synthetic diploid strains were constructed by mating the yeast strain bearing multiple synthetic chromosomes (*synII*, *synIII*, *synVI*, and *synIXR*) to BY4741 and the ring_*synII* synthetic strain to BY4742. The diploid strains were selected and confirmed by polymerase chain reaction (PCR) using specific *MATa/α* primers listed in Supplementary Table 3.

### Plasmid construction
Plasmids used in this study are described in Supplementary Table 2. All plasmids from this study were constructed by standard molecular cloning techniques and Gibson assembly. The EBD domain of the previously developed pSCW11-Cre-EBD plasmid was removed by inverse PCR to generate the pSCW11-Cre plasmid. The site-directed mutagenesis was performed to construct pSCW11 plasmid expressing various Cre variants with an in-frame TAG codon. Primers used in the plasmid construction are listed in Supplementary Table 3.

### GFP-based fluorescence assay
BY4741 strains were co-transformed with the GFP reporter plasmid and the Cre-EBD plasmid or Cre variant plasmids together with the LeuOmeRS/tRNA$_{CUA}$ plasmid. Freshly transformed colonies were isolated and grown in SC−His−Leu−Ura medium at 30 °C for 24 h. Saturated cultures were subcultured into 6 ml of fresh SC−His−Leu−Ura medium (initial OD$_{600}$ = 0.05) supplemented with or without the inducer (1 μM β-estradiol or different concentration of OmeY) and grown to stationary phase at 30 °C for 48 h. All cell cultures were centrifuged at 3000 g for 5 min and resuspended by 2 ml ddH$_2$O. A total of 200 μl resuspending was transferred into a black 96-well plate to measure OD$_{600}$ and green fluorescence ($\lambda_{ex}$ = 488 nm, $\lambda_{em}$ = 520 nm) via a microplate reader (BioTek Synergy H1). Data are reported as the fluorescence intensity divided by the OD$_{600}$ after background subtraction. Yeast cells expressing empty vector and only GFP reporter plasmid were used to determine the background level of this assay. The fluorescence value of yeast cells expressing GFP driven by *ADH1* promoter serve as the denominator to calculate the relative fluorescence intensity.

### GCE-SCRaMbLE
To perform GCE-SCRaMbLE for 1380 isolated SCRaMbLEd strains, synthetic yeast strains with distinct genome ploidy and chromosome conformation were co-transformed with two plasmids, each expressing the Cre variant and the LeuOmeRS/tRNA$_{CUA}$ pair. Freshly

transformed colonies were isolated and grown in SC–His–Leu medium at 30 °C for 12 h to serve as the inoculum. To induce GCE-SCRaMbLE for each designed group, the inoculum was sub-cultured in 30 wells of the 96-well deep-hole plate, each containing 1 ml SC–His–Leu medium (initial $OD_{600} = 0.1$) supplemented with different concentrations of OMeY (see Supplementary Table 2 for more detail). Liquid cultures were aerated by rotary shaking at 220 rpm for 24 h. Every cell culture grown in the 30 different wells of plate (~10,000 yeast cells) was serially diluted and spread onto YPD plates without selection, followed by incubation at 30 °C for 2 days. Two isolated colonies were randomly selected from each plate and grown in 5 ml YPD liquid medium at 30 °C for 3–4 days to prepare the cell cultures for genomic DNA extraction.

To perform GCE-SCRaMbLE for cell pools, the diploid synthetic yeast strains (*synII* and ring_*synII*) co-transformed with plasmids expressing $Cre_{UAG5}$ and $LeuOmeRS/tRNA_{CUA}$ were grown in SC–His–Leu medium at 30 °C for 12 h to serve as the inoculum. Saturated cultures were inoculated into fresh 5 ml SC–His–Leu medium (initial $OD_{600} = 0.1$) supplemented with 1 mM OMeY and grown at 30 °C for 24 h. The cell cultures were directly used for genomic DNA extraction.

## Whole genome sequencing and SCRaMbLE events analysis

Genomic DNA preparation was conducted by using the glass bead method[32]. The sequenced library containing short DNA fragments (200–400 bp) was prepared using the MGIEasy™ Universal DNA Library Prep Kit V1.0 (MGI, Cat number: 1000006985) according to the manual of the protocol. The 100 bp paired-end whole genome sequencing was performed on BGISEQ platform. A total of 500 Mb data was generated for each SCRaMbLEd colony, and 1 Tb data was generated for each SCRaMbLEd cell pools. Reads with unknown bases or low-quality rates (Phred-score < 10) more than 5% were filtered by SOAPnuke[38]. The filtered reads were mapped to reference by SOAPaligner under the default config[39]. The BY4741 served as the reference for the non-synthetic chromosomes available at the *Saccharomyces* Genome Database (http://sgd-archive.yeastgenome.org/sequence/strains/BY4741/BY4741_Toronto_2012/). The reference sequence of synthetic chromosome including *synII*, *synIII*, *synVI*, and *synIXR* are available at GenBank with accession codes CP013608, KC880027, SRX2589074, and JN020955, respectively. To locate the breakpoints involved in SCRaMbLE events, we searched for loxPsym-containing reads, with at least 15 bp of sequence flanking the loxPsym site, that failed to be mapped to the reference. Next, these reads were trisected into a loxPsym site in the middle and two flanking sequences at both ends, which remain associated with the paired sequence in the same DNA fragment. These two flanking sequences were aligned to the reference by Bowtie 2[40] separately to identify novel loxPsym junctions according to the sequence. To prevent false positives, each novel junction was supported by at least 5 splitting reads with the median number at ~20. The reference sequence of each synthetic chromosome was split into several segments according to breakpoints which were then utilized to reconstruct SCRaMbLEd chromosome based on the identification of the parental and novel junctions as well as the copy number of segments flanked by loxPsym sites. The SCRaMbLE event type was then judged by the novel junction and the sequencing depth[12]. Deleted or duplicated segments were estimated by their corresponding sequencing depth. Specifically, duplication and deletion events were identified as repeating regions and regions with a segment copy number of zero in the SCRaMbLEd regions, respectively. Inversion was identified as inversed direction of segments involving two novel junctions.

## HiC-Seq

To prepare the HiC-Seq library, yeast cells were cross-linked for 20 min with 3% formaldehyde at room temperature and quenched with 0.375 M final concentration glycine for 5 min. The cross-linked yeast cells were homogenized by grinding to a fine powder in liquid nitrogen to lyse the cell wall. Endogenous nucleases were inactivated with 0.1% SDS, then chromatin DNA was digested by 100 U *Mbo*I (NEB, Cat number: R0147), and marked with biotin-14-dCTP (Invitrogen, Cat number: 19518018), then ligated by 50 U T4 DNA ligase (NEB, Cat number: M0202). After reversing cross-links, the ligated DNA was extracted through QIAamp DNA Mini Kit (Qiagen, Cat number: 51306) according to manufacturers' instructions. Purified DNA was sheared to 300 to 500 bp fragments and were further blunt-end repaired, A-tailed and adapter-added, followed by purification through biotin-streptavidin–mediated pull-down and PCR amplification[41]. The Hi-C libraries were quantified and sequenced on the MGI-seq platform (BGI, China). The Hi-C sequence data was processed through HiC-Pro to generate normalized matrix at 1 and 10 kb resolution[42]. The 3D coordinate values of contact maps were generated by ShRec3D from the normalized matrix[43]. Then, the coordinate value was translated into PDB format and displayed in PyMOL (version 2.0, Schrödinger, LLC.) as chromosome 3D structures. To better display the number of SCRaMbLE events against the contact counts between the regions involved in recombination events, the scatter plots were treated with a moving average. All data points were first sorted based on their contact counts from the lowest to the highest. The points referring to regions that are not involved in any SCRaMbLE event or inter-chromosomal contact were deleted in the analysis. Based on the aligned points, we then calculated the moving average of SCRaMbLE events and contact counts over a window containing 10 points from top to bottom, shifting the window by three points per calculation. The calculated data points were used to draw the scatter plots.

## Cre enzyme purification

N-terminally His6-tagged Cre and Cre-EBD enzymes were expressed from pET28a plasmids in *E. coli* strain BL21(DE3). The seed culture was grown at 37 °C overnight with shaking at 200 rpm in the Luria-Bertani (LB) medium supplemented with 50 μg/ml Kanamycin (BBI Life Sciences, Cat number: A506636), and was then inoculated into a fresh 200 ml LB medium according to 1% (v/v) dilution factor. The culture flask was grown at 37 °C until the $OD_{600}$ reached 0.5 to 0.6, followed by 1 mM isopropyl-β-d-thiogalactopyranoside (IPTG) (BBI Life Sciences, Cat number: A600168) induction in a shaking incubator (Shanghai Zhichu Instrument Co., Ltd) at 16 °C, 200 rpm for 16 h. Cells pellets were harvested via centrifugation (5000 g for 10 min at 25 °C) for protein purification. The obtained pellets were lysed by the use of a French press (Guangzhou Juneng Nano & Bio-Technology Co., Ltd) (10,000 lb/in²) in 20 ml lysis buffer A (50 mM Tris-HCl [pH 7.4], 33 mM NaCl, 5 mM imidazole, and 5% glycerol) supplemented with 1× protease inhibitor cocktail (Roche, Cat number: 11836170001) and 1 mg/ml DNase I (Tiangen, Cat number: RT411). The cell lysate was clarified by centrifugation (20,000 g for 30 min at 4 °C), and then the supernatant was applied onto the 2 ml nickel resin (ThermoFisher Scientific, Cat number: 88221) equilibrated with 20 ml buffer A. Before the target proteins were eluted with 200 mM imidazole, 30 mM gradient washing was included to remove impurities. Centrifugal filter (10 kDa cutoff, Millipore) was employed for further buffer exchange in buffer B (15 mM Tris-HCl [pH 8.0], 200 mM NaCl, 0.3 mg/ml BSA, 50% glycerol) and stored at −20 °C before use. The purity of purified enzyme was determined in SDS-PAGE followed by Coomassie brilliant blue staining. The amounts of purified enzymes were measured using the Bradford assay kit (Sangon Biotech, Cat Number: C503031).

## In vitro recombination assay of Cre and Cre-EBD enzymes

The activity of purified Cre and Cre-EBD recombinases was measured by the in vitro recombination assay according to a previous study[27]. The PCR-amplified 1432 bp linear DNA containing two direct repeats of the loxP site separated by a ~700 bp spacer was used as the substrate. Fifty microlitres of assay mixture consisted of different amounts of

DNA substrate (120, 240, 360, 480, 600, 720, and 840 ng), 200 ng of the purified Cre or 400 ng Cre-EBD enzymes, and 25 μl buffer C (50 mM Tris·HCl [pH 7.5], 33 mM NaCl and 10 mM MgCl$_2$). The assay mixture was incubated at 37 °C for 30 min and inactivated by incubation at 70 °C for 10 min. 5 μl reaction product was separated on 1% agarose gel, and the band intensities were quantified with ImageJ (version 1.53e). The Michaelis–Menten kinetic analysis was carried out by using the GraphPad Prism (version 9.1.0) to determine the $Km$ and $k_{cat}$ values.

### SDS-PAGE and immunoblotting analysis
Yeast cell pellets (10 OD$_{600}$ units) were resuspended and incubated with 200 μl NaOH (100 mM) at room temperature for 5 min. The NaOH-treated yeast cells were harvested via centrifugation (10,000 g for 2 min) and were subjected to cell lysis treatment via boiling for 10 min in 100 μl of 2x SDS-PAGE loading buffer (Beyotime, Cat number: P0015B). Equivalent levels of protein loading for whole cells were determined on the basis of the OD$_{600}$ of the cell culture (0.1 units per lane) and confirmed by Coomassie's brilliant blue staining of parallel gels. Proteins were separated by reducing 12 % SDS-PAGE and electro-blotted onto polyvinylidene difluoride (PVDF) membranes (Millipore) via eBlotTM L1 (Genscript) transfer system. Cre enzymes were detected by the Cre-specific monoclonal antibody (Invitrogen, Cat number: MA5-27870) and alkaline phosphatase-linked anti-mouse IgG peroxidase antibody (Sigma-Aldrich, Cat number: A2304). Immunoreactive antigens were detected by chemiluminescence using horseradish peroxidase (HRP) substrate (Millipore, Cat number: WBKLS0100).

### Reporting summary
Further information on research design is available in the Nature Research Reporting Summary linked to this article.

## Data availability
The DNA sequencing data of the SCRaMbLEd synthetic yeast strains that support the findings of this study have been deposited in the NCBI database under accession code PRJNA876960 and CNSA (CNGB Nucleotide Sequence Archive) under accession number CNP0002899. The reference genome of BY4741 is downloaded from the *Saccharomyces* Genome Database (http://sgd-archive.yeastgenome.org/sequence/strains/BY4741/BY4741_Toronto_2012/). The reference sequence of *synII*, *synIII*, *synVI*, and *synIXR* used in this study can be downloaded from GenBank with accession codes CP013608, KC880027, SRX2589074, and JN020955, respectively. All data supporting the findings of this study are available within the manuscript file and its Supplementary Information files. Source data are provided with this paper.

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

## Acknowledgements

This work was supported by grants from National Key Research and Development Program of China (No. 2018YFA0900100 to X.F.); National Natural Science Foundation of China (No. 31901029 to X.F.); Natural Science Foundation of Guangdong Province, China (No. 2021A1515010995 to X.F.); Guangdong Provincial Key Laboratory of Genome Read and Write (No. 2017B030301011 to Y.S.). We thank the Sc2.0 consortium for many discussions and collaborations. This work is part of Sc2.0 project (http://syntheticyeast.org/). We are grateful to Prof. James A Van Deventer (Tufts University) for sharing the LeuOmeRS/tRNA$_{CUA}$ plasmid. We are grateful to Dr. Lena Hochrein (University of Potsdam) for sharing the GFP reporter plasmids.

## Author contributions

Y.S. and X.F. conceived the idea and supervised the study. X.F. and Y.S. designed the experiments. H.Z. performed the GCE-SCRaMbLE experiment and data analysis. X.F. and H. Z. constructed the plasmids and performed the fluorescence assay. X.G. performed the Cre purification, in vitro recombination assay, and immunoblotting analysis. Y.W. assisted with sequencing analysis and SCRaMbLEd genome sequence reconstruction. Y.Z. constructed the multiple synthetic yeast strain used in this study. X.F. and Y.S. wrote the manuscript with inputs from all the other authors. All authors commented on the final draft of the manuscript.

## Competing interests

The authors declare the following competing interests: X.F., H.Z., Y.S., and BGI-Shenzhen have filed a patent application (Chinese patent application number: 201910911004.8.) describing the GCE-SCRaMbLE method. The remaining authors declare no competing interests.
