## [Peer Review File · Nature Communications]

Reviewers' Comments:

Reviewer #1:

Remarks to the Author:

Sc 2.0 combined with its built-in feature SCRaMbLE facilitates investigation of fundamental biological questions and yeast engineering for biotechnological applications. Lack of facile and tight regulation of SCRaMbLE however limits the scope of Sc 2.0's utilization. To address this, the authors explored stop codon suppression as a strategy to tightly control Cre expression and thereby SCRaMbLE recombination. In this approach, which is called GCE-SCRaMbLE, full-length expression of Cre is controlled by o-AARS/o-tRNA-mediated incorporation of a non-canonical amino acid (ncAAs) at a TAG stop codon. The authors first identified suitable Cre variants with different TAG positions for stop codon suppression and demonstrate tunable expression dependent on provided concentrations of the ncAA. Then, they subjected 3 haploid and 3 diploid Sc 2.0 yeast strains to different GCE-SCRaMbLE regimes and analyzed derived genome-wide recombination events including insertions, deletions, and duplications.

Overall, this work provides a unique approach to tightly control Cre activity and thereby recombination over a dynamic range. It is impressive that no "leaky" recombination events occurred in the 4 yeast pools tested that harbor the constructs for CreUAG5 or UAG14 and the oAARS/otRNA but are not supplemented with OMeY. The observation that genome ploidy affects deletion events, and that circularization increases the general number of SCRaMbLE events provides valuable insights for future applications of Sc 2.0. In general, I am enthusiastic about the work, but am not an expert in the yeast Sc2.0 project so leave it to members of that community to decide on the level of advance to that field specifically.

Comments:

- 1.) Please, write out abbreviations when using them for the first time in the manuscript. For example, SCRaMbLE or IPTG.
- 2.) Introduction, Results, and Conclusions sections are well written and provide all information needed to understand the work.
- 3.) The Methods section requires some more detail on the instruments used and vendors of chemicals. For example, which shaking incubator and French Press were used for the Cre enzyme purification? who is the vendor for beta-estradiol or OMeY? Please carefully add these details.
- 4.) I struggled understanding the figures easily besides Figure 1. For example Fig 2, according to the scheme in a) there should be 48 test groups, but it's only 23 the authors tested. In addition, what do the colors in the very right figure panel indicate? I believe this could be easily addressed with some more labeling and potentially overviews or schemes under a) illustrating the question which is addressed.
- 5.) Fig 1a) Why do the gene constructs differ in how they are illustrated? Is there a specific reason or could both use the same symbols to depict promoters, terminators, genes? The author show "Relative fluorescence intensities" on the y-axis. Please, provide information about what the reference is in the figure legend.
- 6.) I am not quite understanding why the authors decided for the 24 h induction time. They say that at this time point they observed the highest mortality. According to Suppl. Fig 2, this is true for "ring_synII". The authors also say that they chose this time point because they wanted "to maximize viable deletions". This is confusing to me and sounds like the authors wanted to bias the results towards deletions. Please elaborate to clarify. In addition, it seems that Suppl. Fig 2 misses the error bars. Can you please add them?
- 7.) SCRaMbLE frequencies are depicted as absolute values without standard deviations throughout the manuscript, in particular Fig 2b, c, Fig 3b, d, Suppl. Fig 3, Fig 4. If I understand correctly, the authors picked two clones from the 30 induction per group. I wonder what the observed range is for each. Or are all data summative? It would be informative to understand if there are chromosomes or chromosome regions which are prone to deletion, duplication, inversion, insertion, etc. under all conditions or only under high Cre expression conditions or in haploid strain versus diploid strains.
- 8.) Fig 2c) The number of events cannot be identified for synIII, synVI, synIXR in the way the data are presented. It would help if each chromosome got its own y-axis.
- 9.) Fig 3) I suggest reorganizing b)-e) starting at the upper right, followed by lower left, middle, and right. B) what is the difference between "SCRaMbLE frequency" and "Frequency of events"? C)

Please, write out in the figure legend what exactly is meant by "number of events" – what means events? In addition, I suggest adding the 0 OMeY data. When I first looked through the manuscript, I really missed seeing data for this. You write about it, but it would be helpful to a reader to see in the figures that you performed this important experiment.

10.) Fig. 4) I suggest changing the colors in a) as it is confusing that blue and red mean different things in the same figure.

11.) Fig 5a) please add a y-axis label.

Reviewer #2:

Remarks to the Author:

The manuscript entitled «Systematic dissection of key factors governing recombination outcomes by GCE-SCRaMbLE» propose to explore a well-controlled and tunable expression system for the Cre recombinase in order to perform SCRaMbLE in the Sc2.0 synthetic yeast strains.

The proposed construct and its advantages (lack of leakiness, tunability, ...) compared to previous studies are well described. The panel of analyses and expected results in the various setups (one or several chromosomes, linear or circular chromosomes, haploid or diploid strains) are well illustrated and pave the way for the future exploration of SCRaMbLE in the fully synthetic Sc2.0 yeast strain.

Overall, the paper is well written, and data are presented in a meaningful way. Thus, I think the paper is suitable for publication.

I have a few minor remarks to improve the manuscript:

- It seems to me that there is a small mistake in the boxed legend of Supplementary Figure 4 where the colors for "haploid" and "diploid" strains have been inverted. The written figure legend below being correct.
- Figure 4b: adding "diploid" and "haploid" on their respective sides on the figure itself and not only in the legend would add to the clarity of the figure.
- In the discussion, lines 354 and 360, I believe one should read "circularization" instead of "circulation".
- In the introduction, adding a short explanation on the SCRaMbLE system and what are the loxPsym sites in the synthetic yeast strains would help readers that are unfamiliar with the system to understand it without having to go to previous papers.
- considering the low amount of strains with rearrangements, could you please clarify line 172 if the average number of events per strain has been calculated on all strains or only strains with at least one event.
- line 173, I believe one should read "inter-chromosomal" events rather than "intra-chromosomal".
- In general, these results about the low amount of strains with rearrangements after SCRaMbLE, the low number of events per strain and the absence of translocations across the hundreds of SCRaMbLEd clones stay puzzling to me. They are in discrepancy with the two other papers mentioned as ref 18 and 28, where strains with several synthetic chromosomes were subjected to Cre-EBD under the SCW11 promoter. The question is partly discussed at the end of the manuscript, but I believe it requires deeper thoughts. Indeed, the hypothesis about the low incorporation efficiency of OMeY is a possibility, but hard to reconcile with the GFP-based fluorescence assay in Figure 1b, where the fluorescence intensity recovered after induction with the previously published Cre-EBD is very similar to the one with CreUAG5 described in this paper. Both papers also reported translocations with induction times as low as 2h with the Cre-EBD under the SCW11 promoter and do not state any selection pressure on the SCRaMbLEd strains either. I additionally wonder if no translocations have been found at all, or just no inter-chromosomal events. For instance, are there some events classified as inversion but including the centromere of the respective chromosome? These events could be considered as translocations between chromosome arms instead. Overall, in my opinion, the different behavior of the two Cre constructs in this regard needs further discussion and consideration for future experiments.

We would like to thank the reviewers for their time and suggestions that helped us improve our work. Below, we provide a point-by-point response (in blue color) to reviewers' comments.

Reviewer #1 (Remarks to the Author):

Sc 2.0 combined with its built-in feature SCRaMbLE facilitates investigation of fundamental biological questions and yeast engineering for biotechnological applications. Lack of facile and tight regulation of SCRaMbLE however limits the scope of Sc 2.0's utilization. To address this, the authors explored stop codon suppression as a strategy to tightly control Cre expression and thereby SCRaMbLE recombination. In this approach, which is called GCE-SCRaMbLE, full-length expression of Cre is controlled by o-AARS/o-tRNA-mediated incorporation of a non-canonical amino acid (ncAAs) at a TAG stop codon. The authors first identified suitable Cre variants with different TAG positions for stop codon suppression and demonstrate tunable expression dependent on provided concentrations of the ncAA. Then, they subjected 3 haploid and 3 diploid Sc 2.0 yeast strains to different GCE-SCRaMbLE regimes and analyzed derived genome-wide recombination events including insertions, deletions, and duplications.

Overall, this work provides a unique approach to tightly control Cre activity and thereby recombination over a dynamic range. It is impressive that no "leaky" recombination events occurred in the 4 yeast pools tested that harbor the constructs for CreUAG5 or UAG14 and the oAARS/otRNA but are not supplemented with OMeY. The observation that genome ploidy affects deletion events, and that circularization increases the general number of SCRaMbLE events provides valuable insights for future applications of Sc 2.0. In general, I am enthusiastic about the work, but am not an expert in the yeast Sc2.0 project so leave it to members of that community to decide on the level of advance to that field specifically.

Response 1: We thank the reviewer for recognizing the values of our study and providing very helpful comments.

Comments:

1.) Please, write out abbreviations when using them for the first time in the manuscript. For example, SCRaMbLE or IPTG.

Response 2: We thank the reviewer for this point. All abbreviations are now written out in the updated manuscript highlighted in yellow.

2.) Introduction, Results, and Conclusions sections are well written and provide all information needed to understand the work.

Response 3: We thank the reviewer for the recognition of our work.

3.) The Methods section requires some more detail on the instruments used and vendors of chemicals. For example, which shaking incubator and French Press were used for the Cre enzyme purification? who is the vendor for beta-estradiol or OMeY? Please carefully add these details.

Response 4: We thank the reviewer for this comment. We have included detailed information (highlighted in yellow in the Methods section) about the instruments and vendors of chemicals in the updated methods section.

4.) I struggled understanding the figures easily besides Figure 1. For example, Fig 2, according to the scheme in a) there should be 48 test groups, but it's only 23 the authors tested. In addition, what do the colors in the very right figure panel indicate? I believe this could be easily addressed with some more labeling and potentially overviews or schemes under a) illustrating the question which is addressed.

Response 5: We apologize for the confusion and thank you for the suggestion and comments. Considering the cost, time and efforts of sequencing and whole-genome reconstruction of GCE-SCRaMbLEd yeast strains, we did not test all 48 groups, which requires the WGS analysis of 2880 (=48×30×2) SCRaMbLEd yeast strains. Instead, we specifically designed 23 test groups that are sufficient to fully dissect the influence and corresponding contribution of each factor during the SCRaMbLE process. For instance, we used only Cre_{UAG14} to test the effect of different concentrations of OMeY (1, 2, 5 and 10 mM) on SCRaMbLE outcomes. Please find the design of 23 test groups summarized in Supplemental Table 1. The very right figure panel in Fig. 2A is the representative graphic demonstration of genome rearrangements observed in SCRaMbLEd strains. This form of presentation is the same as used in our previous paper (PMID: 26566658), in which the color of each arrow indicates the different segment flanked by loxPsym recombination sites and the direction of the arrow represents the orientation. We have modified Fig. 2, figure legend and relevant main text to better illustrate our points on page 7 and 26 highlighted in yellow.

5.) Fig 1a) Why do the gene constructs differ in how they are illustrated? Is there a specific reason or could both use the same symbols to depict promoters, terminators, genes? The author show "Relative fluorescence intensities" on the y-axis. Please, provide information about what the reference is in the figure legend.

Response 6: We thank the reviewer for the comments. We have modified Fig. 1A to make the symbols of different gene constructs consistent. The description of the reference to calculate "relative fluorescence intensities" has been included in the figure legend on page 25 highlighted in yellow.

6.) I am not quite understanding why the authors decided for the 24 h induction time. They say that at this time point the observed the highest mortality. According to Suppl.

Fig 2, this is true for “ring_synII”. The authors also say that they chose this time point because they wanted “to maximize viable deletions”. This is confusing to me and sounds like the authors wanted to bias the results towards deletions. Please elaborate to clarify. In addition, it seems that Suppl. Fig 2 misses the error bars. Can you please add them?

Response 7: We thank the reviewer for the comments. SCRaMbLE occurrence rate was found to be positively correlated with cell death rate presumably because recombination events could lead to the deletion of essential genes. Thus, assessment of SCRaMbLE-induced lethality is an easy and quick, although indirect, way to estimate the degree of genome rearrangements induced by SCRaMbLE. Actually, this is the common strategy adopted by others in the Sc2.0 consortium (PMID: 29789540, 29789590). To acquire strains with SCRaMbLE events as much as possible when no selection was introduced, induction time at 24-hour was chosen as cells exhibited the highest mortality. The main text is modified to better illustrate our point on page 7 highlighted in yellow. In addition, we repeated the experiments of Suppl. Fig 2 and the error bars are now added in the updated Suppl. Fig 2.

7.) SCRaMbLE frequencies are depicted as absolute values without standard deviations throughout the manuscript, in particular Fig 2b, c, Fig 3b, d, Suppl. Fig 3, Fig 4. If I understand correctly, the authors picked two clones from the 30 induction per group. I wonder what the observed range is for each. Or are all data summative? It would be informative to understand if there are chromosomes or chromosome regions which are prone to deletion, duplication, inversion, insertion, etc. under all conditions or only under high Cre expression conditions or in haploid strain versus diploid strains.

Response 8: We thank the reviewer for the comments. The data used for plotting Fig 2b, c, and Suppl. Fig 3 are the absolute number of SCRaMbLE events. The data used for plotting Fig 3b, d is the absolute number of strains with SCRaMbLE events in each group (N=180). Each dataset used for analysis is unique. Thus, standard deviations are not applicable to these figures. We modified the y-axis in the updated Fig. 2b and Fig. 3b, d to avoid confusion.

As mentioned by the reviewer, the data shown in Fig. 4a and Supplementary Fig. 4 are summative, because we simply wanted to show the effect of genome ploidy on deletion capability via SCRaMbLE. We believe this form of presentation could clearly illustrate our point that the number of strains with deletion and the degree of chromosome loss was significantly higher in the diploid strains than haploid strains. In addition, we did not observe apparent effect of different Cre expression on deletion capability by GCE-SCRaMbLE (Supplementary Fig. 4). In the Fig. 4b, we further generated the SCRaMbLE rearrangement landscape that provides key information about *synII* chromosome regions which are prone to deletion, duplication, inversion. We have added more descriptions of relevant findings in the main text according to the reviewer's suggestions on page 10 highlighted in yellow.

8.) Fig 2c) The number of events cannot be identified for *synIII*, *synVI*, *synIXR* in the way the data are presented. It would help if each chromosome got its own y-axis.

Response 9: We thank the reviewer for this point. The y-axis for each chromosome is now added in the updated Fig. 2C.

9.) Fig 3) I suggest reorganizing b)-e) starting at the upper right, followed by lower left, middle, and right. B) what is the difference between “SCRaMbLE frequency” and “Frequency of events”? C) Please, write out in the figure legend what exactly is meant by “number of events” – what means events? In addition, I suggest adding the 0 OMeY data. When I first looked through the manuscript, I really missed seeing data for this. You write about it, but it would be helpful to a reader to see in the figures that you performed this important experiment.

Response 10: We thank the reviewer for the comments. The layout of the Fig.3 is modified according to the suggestion.

“SCRaMbLE frequency” means the SCRaMbLE occurrence rate within a post-SCRaMbLE yeast population (N=180 colonies). We changed the y-axis of Fig. 3b, d from “SCRaMbLE frequency” to “SCRaMbLE occurrence rate in population” to avoid confusion. “Frequency of events” of Fig. 2b means the proportion of different types of SCRaMbLE events in all SCRaMbLE events. Y-axis of Fig. 2b is also modified to avoid confusion.

The “number of events” in Fig.3 means the total number of all types of SCRaMbLE events including deletion, duplication, and inversion. The Fig.3 legend is modified to better clarify this on page 27 highlighted in yellow.

We did not perform whole genome reconstruction and analysis of isolated SCRaMbLEd strains subjected to GCE-SCRaMbLE in the absence of OMeY, since we expected to see no rearrangement events. Instead, four yeast pools (haploid *syn2369R* and *ring_synII* strains expressing *Cre_{UAG5}* and *Cre_{UAG14}* respectively) in the absence of OMeY were prepared for deep sequencing (~17,000x) to identify potential “leaky” events that might be overlooked by single colony selection. Thus, 0 OMeY data is not applicable to Fig. 3.

10.) Fig. 4) I suggest changing the colors in a) as it is confusing that blue and red mean different things in the same figure.

Response 11: We thank the reviewer for this comment. The colors of panel A in Fig.4 is now modified.

11.) Fig 5a) please add a y-axis label.

Response 12: We thank the reviewer for this comment. The y-axis label is added.

Reviewer #2 (Remarks to the Author):

The manuscript entitled «Systematic dissection of key factors governing recombination outcomes by GCE-SCRaMbLE» propose to explore a well-controlled and tunable expression system for the Cre recombinase in order to perform SCRaMbLE in the Sc2.0 synthetic yeast strains.

The proposed construct and its advantages (lack of leakiness, tunability, ...) compared to previous studies are well described. The panel of analyses and expected results in the various setups (one or several chromosomes, linear or circular chromosomes, haploid or diploid strains) are well illustrated and pave the way for the future exploration of SCRaMbLE in the fully synthetic Sc2.0 yeast strain.

Overall, the paper is well written, and data are presented in a meaningful way. Thus, I think the paper is suitable for publication.

Response 13: We thank the reviewer for recognizing the values and significance of our study and for providing insightful comments.

I have a few minor remarks to improve the manuscript:

- It seems to me that there is a small mistake in the boxed legend of Supplementary Figure 4 where the colors for “haploid” and “diploid” strains have been inverted. The written figure legend below being correct.

Response 14: We thank the reviewer for pointing out this mistake. We have corrected Supplementary Figure 4 in the updated manuscript.

- Figure 4b: adding “diploid” and “haploid” on their respective sides on the figure itself and not only in the legend would add to the clarity of the figure.

Response 15: We thank the reviewer for this comment. Fig. 4b has been modified according to the suggestion.

- In the discussion, lines 354 and 360, I believe one should read “circularization” instead of “circulation”.

Response 16: We thank the reviewer for pointing out this mistake. We have corrected in the updated manuscript.

- In the introduction, adding a short explanation on the SCRaMbLE system and what are the loxPsym sites in the synthetic yeast strains would help readers that are unfamiliar with the system to understand it without having to go to previous papers.

Response 17: We thank the reviewer for this comment. The first paragraph of introduction is now expanded to further explain about the SCRaMbLE system.

- considering the low amount of strains with rearrangements, could you please clarify line 172 if the average number of events per strain has been calculated on all strains or only strains with at least one event.

Response 18: We thank the reviewer for this comment. An average number of SCRaMbLE events per strain at around 4 was observed in strains with at least one event. The main text is rewritten to better clarify this point on page 7 highlighted in yellow.

- line 173, I believe one should read “inter-chromosomal” events rather than “intra-chromosomal”.

Response 19: We thank the reviewer for pointing out this mistake. We have corrected in the updated manuscript.

- In general, these results about the low amount of strains with rearrangements after SCRaMbLE, the low number of events per strain and the absence of translocations across the hundreds of SCRaMbLEd clones stay puzzling to me. They are in discrepancy with the two other papers mentioned as ref 18 and 28, where strains with several synthetic chromosomes were subjected to Cre-EBD under the SCW11 promoter. The question is partly discussed at the end of the manuscript, but I believe it requires deeper thoughts. Indeed, the hypothesis about the low incorporation efficiency of OMeY is a possibility, but hard to reconcile with the GFP-based fluorescence assay in Figure 1b, where the fluorescence intensity recovered after induction with the previously published Cre-EBD is very similar to the one with CreUAG5 described in this paper. Both papers also reported translocations with induction times as low as 2h with the Cre-EBD under the SCW11 promoter and do not state any selection pressure on the SCRaMbLEd strains either. I additionally wonder if no translocations have been found at all, or just no inter-chromosomal events. For instance, are there some events classified as inversion but including the centromere of the respective chromosome? These events could be considered as translocations between chromosome arms instead. Overall, in my opinion, the different behavior of the two Cre constructs in this regard needs further discussion and consideration for future experiments.

Response 20: We thank the reviewer for the insightful comments. To the best of our knowledge, the SCRaMbLE occurrence rate in the population scale without selection pressure has not been systematically investigated previously. Although the work presented in reference 28 (PMID: 28280150) provides some useful information, it still is not enough to answer the question (eg: reference 28 does not specifically mention the two clones used for analysis were picked out from how many independent clones and whether SCRaMbLE events were identified in other clones or not). We think another relevant study for this topic is the reference (PMID: 29789540); its Fig. 4 shows the proportion of SCRaMbLEd cells with altered fluorescence value, an indicator for

strains with rearrangements, is significantly lower for random sampling compared to selective sampling. These findings suggest the relatively low amount of strains with rearrangements after SCRaMbLE when no selection pressure was applied, although this assay is based on indirect evidence and could not truly estimate the SCRaMbLE occurrence rate in the population. To acquire direct and systematic information, our study specifically designed the experiment to dissect SCRaMbLE occurrence rate in the population and the number of events per strain by sequencing more than one thousand randomly picked clones subjected to GCE-SCRaMbLE.

To answer the reviewer's question that why the number of SCRaMbLE events in our study is much less than that shown in another unpublished work of reference 18 on the bioRxiv (<https://doi.org/10.1101/2021.07.19.453002>), we further compared the analysis workflow. The main differences include two parts: First and most importantly, identifying a rearrangement event would require at least two essential information: the identification of novel junctions and the depth of corresponding segments (the detailed analysis pipeline is described in our previous work, PMID: 26566658). Using pool sequencing of a population subjected to SCRaMbLE presented in their work can provide junction information, but fail to provide depth information of the segment per strain. It is worth pointing out that ref 18 could not determine the occurrence rate of SCRaMbLE in this population because the number of SCRaMbLEd clones remains unknown. In our study, we performed whole genome reconstruction of all isolated SCRaMbLEd clones, so we were able to confidently identify the SCRaMbLE events in each strain. Second, the supporting reads number for a novel junction in our study is set higher (≥ 5 with the median number around 20) than that in ref 18 (≥ 2) to avoid false positive results. Thus, we think the number of SCRaMbLE events in their study is likely overestimated. Overall, we believe that our work, for the first time, has shed light on SCRaMbLE occurrence rate in the population and the number of events per strain without selection pressure. The second paragraph in the discussion section is rewritten and expanded in the updated manuscript on page 13-14.

To answer the reviewer's question that why no translocation was observed across the hundreds of SCRaMbLEd clones in our study: In the previous version of the manuscript, we only showed SCRaMbLE event types including inversion, deletion, and duplications because they are the most basic type of event that can be identified directly by novel junctions and depth of segments according to our previously established analysis pipeline (PMID: 26566658). Translocation would require further manual judgment by combining two-step events (one loop-out and one following insertion). To follow the reviewer's suggestion, we carefully analyzed our results by involving manual judgments and now discovered 4 translocation events with the representative diagram shown below (right panel). In our previous analysis, these 4 translocation events were identified as 8 inversion events because two consecutive inversion events could also give rise to the same rearranged chromosome (left panel of the figure below). Thus, in the updated main text we divided SCRaMbLE events into four types including deletions, inversions, duplications and translocations with the modified number of events (page

8 highlighted in yellow). Fig. 2b and Fig. 4b are also modified accordingly.

Reviewers' Comments:

Reviewer #1:

Remarks to the Author:

The authors have addressed my comments and I believe the manuscript is strong.

Reviewer #2:

Remarks to the Author:

I thank for the authors for following advice and the general clarification of the text and figures according to my comments and the ones from the other reviewer.

I want to come back on the main issue that I raised about the absence of translocation found in this dataset. I thank the authors for giving in depth explanation of their findings and classification of events in the reviewers' comments. In particular, the graphical explanation of their new classification allowed me to understand that there is a major misunderstanding on the meaning of "translocation". By translocation, I do not mean a central portion of a chromosome "jumping" to another location on that same chromosome. I totally agree that these events can be either explained by two consecutive inversions or by a deletion/circularization followed by re-insertion at a different location. I do not know how they should be called and whether they have to be considered separately, but they do not correspond to translocations. When talking about genome rearrangements in general, people refer to translocation as a move of a full end of a chromosome arm to another. They are usually depicted involving two different chromosomes but can be two arms of the same chromosome, thus looking exactly as a large inversion including the chromosome centromere (pericentric inversion). They can also be (and often are) reciprocal, which avoids any loss of large portion of the genome, which would in general be very deleterious if not lethal. Depending on the event, these could then correspond to only one, if just a simple (non-reciprocal) translocation, but most often two new junctions. So, from my understanding, the authors found none of these types of events amongst the huge amount of analyzed clones, which is different from the other studies -in which I agree it is impossible to state any frequency of events- where several translocations have been found in a probable much smaller sampling. I included a small schematic as additional file to clarify this point.

In light of this clarification, I hope the authors can then correct their current version of the paper (probably reverting fig 2b back to the previous version) and discuss the absence of inter-chromosomal/translocation events in the manuscript.

We would like to thank the reviewers for their time and suggestions that helped us improve our work. Below, we provide a point-by-point response (in blue color) to reviewers' comments.

Reviewer #1 (Remarks to the Author):

The authors have addressed my comments and I believe the manuscript is strong.

Response: We thank the reviewer for recognizing the value of our study.

Reviewer #2 (Remarks to the Author):

I thank for the authors for following advice and the general clarification of the text and figures according to my comments and the ones from the other reviewer.

I want to come back on the main issue that I raised about the absence of translocation found in this dataset. I thank the authors for giving in depth explanation of their findings and classification of events in the reviewers' comments. In particular, the graphical explanation of their new classification allowed me to understand that there is a major misunderstanding on the meaning of "translocation". By translocation, I do not mean a central portion of a chromosome "jumping" to another location on that same chromosome. I totally agree that these events can be either explained by two consecutive inversions or by a deletion/circularization followed by re-insertion at a different location. I do not know how they should be called and whether they have to be considered separately, but they do not correspond to translocations. When talking about genome rearrangements in general, people refer to translocation as a move of a full end of a chromosome arm to another. They are usually depicted involving two different chromosomes but can be two arms of the same chromosome, thus looking exactly as a large inversion including the chromosome centromere (pericentric inversion). They can also be (and often are) reciprocal, which avoids any loss of large portion of the genome, which would in general be very deleterious if not lethal. Depending on the event, these could then correspond to only one, if just a simple (non-reciprocal) translocation, but most often two new junctions. So, from my understanding, the authors found none of these types of events amongst the huge amount of analyzed clones, which is different from the other studies -in which I agree it is impossible to state any frequency of events- where several translocations have been found in a probable much smaller sampling. I included a small schematic as additional file to clarify this point.

In light of this clarification, I hope the authors can then correct their current version of the paper (probably reverting fig 2b back to the previous version) and discuss the absence of inter-chromosomal/translocation events in the manuscript.

Response: We thank the reviewer for the detailed clarification of the definition of translocation. Following the reviewer's suggestion, we reverted fig 2b and fig 4b back to the previous version and modified the relevant main text accordingly as highlighted

in the manuscript. We also added some discussions about the absence of inter-chromosomal/translocation events in the manuscript (on page 14 highlighted in yellow).